# The Hydrolytic Activity of Copper(II) Complexes with 1,4,7-Triazacyclononane Derivatives for the Hydrolysis of Phosphate Diesters

**DOI:** 10.3390/molecules28227542

**Published:** 2023-11-11

**Authors:** Michaela Buziková, Robert Willimetz, Jan Kotek

**Affiliations:** Department of Inorganic Chemistry, Charles University, Hlavova 2030, 128 40 Prague, Czech Republic

**Keywords:** artificial nuclease, coordination compound, bifunctional ligand

## Abstract

A set of substituted 1,4,7-triazacyclononane ligands was synthesised, including a wide series of novel derivatives bearing a thiazole or thiophene side group, with the potential to incorporate these derivatives into a polymeric material; some previously known/studied ligands were also synthesised for comparative purposes. The corresponding copper(II) complexes were prepared, and their ability to mediate the hydrolysis of phosphate ester bonds was studied via UV-Vis spectrophotometry, using bis(*p*-nitrophenyl)phosphate as a model substrate. Some of the prepared complexes showed a considerable enhancement of the phosphate ester hydrolysis in comparison with previously studied systems, which makes them some of the most effective complexes ever tested for this purpose. Therefore, these novel, potentially bifunctional systems could provide the possibility of creating new coating materials for medicinal devices that could prevent biofilm formation.

## 1. Introduction

Biofilms present a significant problem in current medicine, mainly due to their ability to colonise almost any surface and their extensive antibiotic resistance [1]. Through a polymeric substance called a matrix, bacteria adhere to the surfaces of catheters, cannulas or stents, forming biofilms and therefore causing infections and chronic wounds which are difficult to heal. The main components of the matrix are proteins and extracellular DNA (eDNA). A possible solution for preventing biofilm formation seems to be the catalytic degradation of eDNA using artificial nucleases, compounds which are able to cleave the stable phosphodiester bonds in the DNA backbone. A number of metal complexes were tested for this purpose, namely those of transition metal ions such as Cu(II), Ni(II), Zn(II), Co(III) and Fe(II) or of lanthanides [2,3,4,5]. In particular, coordinatively unsaturated Cu(II) complexes of 1,4,7-triazacyclononane (TACN) derivatives have been shown to present significant hydrolytic activity [2,3,6,7].

Copper(II) ions are well known to form very stable complexes with TACN itself [8,9] and its derivatives, especially if they contain coordinating pendant arms, although a distorted coordination sphere is formed around the central ion with coordination numbers 5–6 due to the Jahn–Teller phenomena [10]. If coordinating pendant arms are not present, the copper(II) ion is facially coordinated by the TACN skeleton, which leaves 2–3 coordination sites free for the coordination of other ligand(s). According to this fact, a mechanism of hydrolysis of the phosphate ester group was suggested, involving pentacoordinated species in which water/hydroxido/phosphato ligands occupy the free positions of the Cu(II)–TACN complex [11,12,13]. This generally accepted mechanism is shown in Figure 1.

It was shown that the existence of monomer–dimer equilibrium limits the reactivity of Cu(II)–TACN and other related complexes as the dimer and di-hydroxido species are inactive in the catalysis of the hydrolysis [11,12,13]. It was found that the introduction of sterically bulky substituents on the TACN skeleton renders the dimerisation less favourable, therefore enhancing the observed reaction rates of the hydrolysis [11,12,13,14,15]. The introduction of only two substituents is sufficient to significantly enhance reactivity in comparison with the parent Cu(II)–TACN complex [15]. On the other hand, the presence of potentially coordinating group(s) close to the TACN core of the ligand can significantly reduce the reactivity as the potential formation of one or more additional stable chelate rings reduces the number of coordination sites on the Cu(II) centre. It was demonstrated that primary amino groups present in the pendant substituents completely inhibited the complex’s hydrolytic activity [15].

For the possible utilisation of an artificial nuclease system on a selected surface, the bifunctionalisation of the ligand is needed—in addition to a metal-binding ability, the ligand must contain a suitable side-group which serves to anchor the ligand (and/or its complex) onto the given surface via chemisorption or co-polymerisation into a polymer coating layer. Based on the anchoring group’s nature, it can be potentially coordinated to the coordinatively unsaturated metal ion in the Cu(II)–TACN moiety, which can block the active site and prevent the coordination and hydrolysis of the substrate. Therefore, the linker’s length can also be important [16].

In this study, we report on the synthesis and hydrolytic activity of copper(II) complexes with bifunctional systems designed for the hydrolysis of a phosphodiester moiety. A ligand series was chosen to study the influence of the nature of the anchoring group and the linker’s length on the hydrolytic activity. Structural formulas of the potentially bifunctional designed ligands are shown in Figure 1. 1,4-di(*iso*-propyl)-TACN was chosen as the ligating unit due to promising data in the literature [12]. The ligating unit was substituted with an amino anchor (**L10a**–**c**) suitable for amidic coupling and thiazole (**L11a**–**c**, **L12a**,**b**) and thiophene (**L13a**–**c**, **L14a**–**c**) moieties, which would enable its incorporation into pyrrole or thiophene polymers (e.g., PEDOT) [17,18]. Simple aliphatic chains of different lengths (from C_3_ to C_5_, **L11a**–**c**, **L13a**–**c** and **L14a**–**c**) and acetamide/propionamide groups (**L12a**,**b**) were chosen as linkers. The thiazole/thiophene anchor was connected to the ligating TACN system through the amidic coupling of thiazole-4-carboxylic acid/thiophene-3-carboxylix acid with the amino group terminating an aliphatic chain bound to the ligating part of the TACN (**L11a**–**c**, **L12a**,**b**, **L13a**–**c**) or directly as a (thiophene-3-yl)alkyl substituent (**L14a**–**c**). Although a significant amount of data on the hydrolase activity of Cu(II) complexes with simple TACN derivatives has been reported in the literature (see e.g., [3]), the data for individual systems were, in some cases, acquired under very different conditions, and the reported values are thus often not directly comparable with new results. Therefore, we also studied a series of simple ligands, **L1**–**L9** (Figure 1), to obtain their hydrolytic activity under the same conditions determined for our target (potentially bifunctional) compounds, **L10**–**L14**.

For this work, bis(4-nitrophenyl) phosphate (BNPP) was chosen as a model substrate. This compound has been used as a standard for a very long time by several authors [6] as its hydrolysis can be simply determined via UV-Vis spectrophotometry due to the formation of 4-nitrophenolate ion which is intensively yellow at a neutral-to-alkaline pH (NP; Figure 2) and has a very strong absorption band at around 400 nm. It was shown that the hydrolytic efficiency found in the experiments with BNPP can be correlated with the hydrolysis of real DNA samples [19].

## 2. Results

### 2.1. Ligand Synthesis

Ligands **L2** [20] and **L3** [21] were prepared using established methodology. The preparation of the symmetrically trisubstituted ligands **L6** and **L7** began using commercially available TACN (**L1**); methyl groups were introduced via the Eschweiler–Clarke reaction, whereas *iso*-propyl groups were introduced via alkylation, employing *iso*-propyl bromide (Figure 3). An analogous methodology was used for the preparation of ligands **L8** and **L9,** starting from **L3** (Figure 4). The benzyl group present in compounds **L8** and **L9** was removed via catalytic hydrogenation, affording ligands **L4** and **L5**, respectively (Figure 4).

The synthesis of the potentially bifunctional ligands **L10a**–**c** and **L11a**–**c** began with the di(*iso*-propyl) derivative **L5** (Figure 5). In the first step, phthaloyl-protected ω-aminoalkyl substituent was introduced via alkylation with the corresponding bromo derivative **1a**–**c**. Then, the phthaloyl-protecting group in compounds **2a–c** was removed via hydrazinolysis, affording compounds **L10a**–**c** which contain a primary amino group. A thiazole moiety was then introduced via the amidic coupling of the primary amino group with thiazole-4-carboxylic acid (**3**), using hexafluorophosphate azabenzotriazole tetramethyl uronium (HATU) as a coupling agent. However, this approach afforded the final bifunctional ligands **L11a**–**c** in low yields, and extensive chromatography was needed during the processing of the reaction mixtures. An alternative amide-coupling pathway employing thiazole-4-carbonyl chloride (freshly formed by reacting thiazole-4-carboxylic acid with SOCl_2_) produced no or negligible yields of **L11a**–**c**, probably due to the low solubility of the acylation agent, which was formed in the form of hydrochloride, in aprotic solvents. The use of 1-ethyl-3-(3-dimethylaminopropyl)carbodiimide (EDC) as the coupling agent resulted in no formation of **L11a**–**c**.

Ligands **L12a**,**b** with an amide linker were prepared via the alkylation of di(*iso*-propyl) derivative **L5** with appropriate pendant arm precursor **6a**,**b** (Figure 6). The pendant arm precursors **6a**,**b** were prepared via the amidation of methyl-4-thiazolecarboxylate (**4**) with an excess of ethylenediamine and the acylation of the remaining primary amino group in the intermediate **5** with chloroacetic acid chloride or chloropropionic acid chloride, respectively.

Ligands **L13a**–**c** were prepared according to Figure 7 via the amidic coupling of the corresponding ligands **L10a**–**c** with thiophene-3-carbonyl chloride (**7**).

Ligands **L14a**–**c** were prepared according to Figure 8. In the first step, Boc-protected precursor **8** was alkylated with a slight excess of the corresponding 3-(bromoalkyl)thiophene **9a**–**c** to obtain the protected intermediate **10a**–**c**, which was subsequently deprotected by aq. HCl, giving monosubstituted TACN derivatives **11a**–**c**. The secondary amino groups in **11a**–**c** were alkylated with *i*-PrBr, affording the required ligands **L14a**–**c**.

### 2.2. The Hydrolysis of BNPP Mediated by Cu(II) Complexes

Cu(II) complexes of the studied ligands were prepared in an aqueous solution, and the pH of the solutions was adjusted to 8.5. The complexation reactions were very fast, as documented via a fast change in colour from light blue (Cu(II) aqua-ion in an initial CuCl_2_∙2H_2_O or Cu(OAc)_2_ solution) to dark blue or green (Cu(II)–**L** complex). The corresponding ligand was used in a slight excess (~10%) to assure the full complexation of the metal ion; when an excess of Cu(II) was accidentally present in the solution, it led to the precipitation of colloidal Cu(OH)_2_ during pH adjustment. Unfortunately, the Cu(II)-complex of **L7** was not soluble in water; thus, its hydrolytic activity could not be studied under the selected conditions. The long-term stability of prepared stock solutions of the complexes was tested via UV-Vis spectroscopy; the absorption spectra remained the same at least for several months, and no precipitate was formed. All complexes showed similar spectra in the visible region—a broad d-d band with an absorption maximum at ca. 630–680 nm (Appendix A).

The qualitative identification of NP and NPP suggested in Figure 1 as products of the hydrolysis of BNPP was performed via HPLC. To supress the salinity of the sample to conduct the HPLC, NH_4_HCO_3_ was used as a buffer in the hydrolytic reaction as it can be easily removed during the pre-treatment of the sample for HPLC. The Cu(II)–**L1** complex strongly absorbs in UV region; therefore, it needed to be removed before the HPLC was conducted. The removal of the complex (together with the buffer) was carried out via chromatography on a strong cation exchange resin, and a mixture of BNPP, NPP and NP eluted with water was directly analysed. An example of the HPLC carried out is shown in Figure 2. With time, the peak area of the BNPP gradually decreased, whereas the peak areas of NPP and NP increased.

The hydrolytic activity of the studied complexes (Figure 1) was monitored in a spectral range of approx. 350–450 nm in which the absorption band of the NP gradually increased. A large (100:1) excess of the Cu(II) complex (5 mM) over the BNPP (0.05 mM) was used as these conditions have been commonly used in the past, as reported in the literature [6,7,11]. As an example, the change in the spectra of the Cu(II)–**L13a**–BNPP system is shown in Figure 3 with the time dependence of the absorbance in the absorption maxima of the NP ion; analogous data obtained for other systems are shown in ESI. The values of *k*_obs_ obtained for the studied set of complexes are listed in Table 1.

In addition to the significant increase in absorbance at 400 nm due to the formation of NP, a very small spectral change in the d-d band can also be seen (for example, see the inset in Figure 3a); it is a tiny systematic increase in the absorption with time. It can be attributed to the gradual binding of the NPP to the free coordination site of the Cu(II)–**L** complex as the concentration of NPP increases during the course of the hydrolytic reaction and also because NPP is a better donor than BNPP due to its higher overall negative charge. Such coordination slightly changes the overall ligand field and leads to a small change in the absorption coefficient. However, it can also potentially block the metal centre from mediating the hydrolytic reaction (i.e., an increase in the concentration of NPP leads to catalyst poisoning). To test this hypothesis, we performed the hydrolytic reaction with a mixture of BNPP:Cu(complex) at a 10:1 ratio (*c*(Cu(II) complex) = 0.01 mM; *c*(BNPP) = 0.1 mM)]. The Cu(II)–**L14b** system was used for this study as it displayed the fastest hydrolysis in the previous experiment (see Table 1). If the hydrolytic reaction was catalytic, as expected based on [11,12,13] and reflected in Figure 1, the concentration of NP should increase almost linearly with time. However, a bended, saturation-like trend was observed, with a significant curvature of the dependence corresponding to the hydrolysis of ca. 1 equiv. of BNPP (Figure 4). Therefore, the hydrolytic reaction was significantly slowed by increasing the concentration of the product yet still proceeded in a catalytic fashion.

## 3. Discussion

Three premises can be formulated based on data from the literature [11,12,13,14,15]: (i) the presence of some non-coordinating substituents on the amino groups of the TACN scaffold increases the hydrolytic activity of the complexes; (ii) the bulkier the substituents on TACN, the higher the observed rate constant of the hydrolytic reaction; and (iii) the presence of coordinating group(s) in the vicinity of the macrocyclic unit nullifies the catalytic ability of the copper ion. As can be seen from Table 1, assumption (i) can be nicely demonstrated using the series of complexes of the ligands **L1**, **L2**, **L4** and **L6** for which the observed rate constant rises gradually with the increasing number of methyl substituents bound to the TACN. Assumption (ii) seems to be proven by comparing the efficiency of the complexes of **L4** and **L5**. On the other hand, this suggestion is violated when comparing the systems with **L8** and **L9**. Here, the substituents on ligand **L9** are probably too bulky to allow for an easy approach of the substrate, which is reflected in the lowering of the observed rate constant. Considering the proposed mechanism of phosphate hydrolysis via Cu(II)–TACN complexes shown in Figure 1, hydrolysis cannot proceed if some of the coordination sites of the metal are occupied by a coordinating group present in the ligand in “the close vicinity” of the macrocyclic centre, which led to assumption (iii). Comparing the data acquired for the complexes of **L10a**–**c**, it is clear that the aminopropyl substituent in **L10a** (with the potential to coordinate the Cu(II) ion via the formation of a six-membered chelate ring) efficiently blocks the hydrolysis, consistent with the previous finding reported for a related bis(3-aminopropyl) derivative [15]. However, after increasing the spacer length to 1,4-butylene or 1,5-pentylene, very efficient systems were obtained with ligands **L10b**,**c**, and the efficiency increased with the length of the spacer, as expected (Table 1). When the terminal amino group in **L10a**–**c** is substituted with thiophenecarbonyl group, the blocking ability of the amino group in **L10a** is decreased due to amide formation (as the amide nitrogen atom cannot be easily coordinated to the metal centre), and the complexes of all three ligands **L13a**–**c** have comparable efficiencies which slightly increase with the spacer length. However, analogous thiazolecarboxy derivatives **L11a**–**c** have been found to be completely ineffective in catalytic hydrolysis, and only negligible hydrolytic activity was found for some of them when the reaction mixture was heated up to 50 °C (Table 1). In these cases, a thiazolecarboxamide moiety, which can coordinate the Cu(II) ion via the formation of five-membered N–O and N–N chelate rings, obviously blocks the catalytically active centre in the complexes. The complexes of **L12a**,**b** with acetamide and propionic amide pendant arms do not catalyse the hydrolytic reaction at all—the oxygen atom of the amide group obviously strongly coordinates the central ion. On the contrary, the systems with ligands **L14a**–**c** with thiophene–alkyl anchors were found to be the most efficient, although the efficiency of the Cu(II)–**L14c** complex is slightly decreased because of its low solubility in aqueous media (Table 1).

Although the studied hydrolytic reaction is generally considered to be catalytic [11,12,13] as reflected in Figure 1, the observed significant lowering of the reaction rate with time when the hydrolytic reaction was performed at a 10:1 ratio of BNPP:Cu(II)–**L14b** shows that the hydrolysis product, NPP, probably binds to the metal centre significantly more strongly than the BNPP substrate and acts as a catalyst poison (Figure 4). However, if the complex is expected to be used under heterogeneous conditions, the reaction rate should stay high if the NPP formed via hydrolysis is removed (e.g., by washing the material or via the circulation of bodily fluids).

## 4. Materials and Methods

### 4.1. General

Commercial chemicals (Fluka (Dresden, Germany), Aldrich (St. Louis, MO, USA) and Lachema (Brno, Czech Republic)) were used as received. Anhydrous solvents were obtained via established procedures [22] or purchased. 3-Bromopropyl-phthalimide and 4-bromobutyl-phthalimide **1a**,**b** were prepared according to [23], and 5-bromopentyl-phthalimide **1c** was prepared analogously using 1,5-dibromopentane and was and purified via silica chromatography using 2:1 hexane/ethyl acetate as a mobile phase. 1,4-Bis(*tert*-butyloxycarbonyl)-1,4,7-triazacyclonane (**8**) was prepared using a modified procedure compared to the previously published procedure, with a slightly improved yield [24]. 4-Thiazolecarboxylic acid (**3**) [25], 3-bromothiophene [26] and 3-(4-bromobutyl)thiophene (**9b**) [27] were prepared according to procedures in the literature; the 3-(3-bromopropyl) and 3-(5-bromopentyl) analogues **9a**,**c** were prepared analogously. 3-Thiophenecarboxylic acid was prepared via bubbling gaseous CO_2_ through a solution of thiophene-3-yl-magnesium bromide [28] in Et_2_O. 4-Thiazolecarbonyl chloride and 3-thiophenecarbonyl chloride (**7**) were prepared via refluxing the corresponding acid in SOCl_2_ and evaporation. Methyl-4-thiazolecarboxylate (**4**) was prepared via refluxing 3-thiazolecarbonyl chloride in MeOH and evaporation.

1,4,7-triazacyclononane (TACN, **L1**) was purchased from CheMatech. 1-methyl-1,4,7-triazacyclononane (**L2**) [20] and 1-benzyl-1,4,7-triazacyclononane (**L3**) [21] were prepared using reported procedures. Tris(4-nitrophenyl) phosphate was prepared via the reaction of POCl_3_ with 3 equiv. of 4-nitrophenol in acetonitrile in presence of 3 equiv. of Et_3_N. Bis(4-nitrophenyl) phosphate was prepared via the partial hydrolysis of tris(4-nitrophenyl) phosphate with LiOH in 1,4-dioxane and was recrystallised from water [29].

Thin-layer chromatography (TLC) was performed on silica-coated aluminium sheets (Silica gel 60 F254 (Merck)). Spots were visualised using UV light (254 nm), dipping the sheets in either a 0.5% ethanolic solution of ninhydrin, 5% aq. CuSO_4_, Dragendorff reagent, 10% phosphomolybdenic acid in EtOH, 1% aq. KMnO_4_/5% aq. Na_2_CO_3_ or 1% vanillin in EtOH.

FLASH chromatography was performed using an ECOM ECS28P0X (UV-Vis diode array detector, *λ* = 200–800 nm). Reverse-phase chromatography was carried out using 40 g columns prefilled with C18 stationary phase (YMC-Dispopack AT, ODS-25, Lot: 13578). Methods employing a gradient of MeCN in H_2_O (both solvents with 1 ‰ of TFA) at a flow rate of 50 mL/min were used. General method 1 (GM1): 0.0–2.0 min 20% MeCN; 2.0–15.0 min linear gradient from 20% MeCN to 100% MeCN, 15.0–18.0 min 100% MeCN. Alternatively, general method 2 (GM2): 0.0–2.0 min 5% MeCN; 2.0–10.0 min linear gradient from 5% MeCN to 100% MeCN, 10.0–12.0 min 100% MeCN.

NMR spectra were recorded using NMR spectrometers, a Varian VNMRS300 (300 MHz for ^1^H and 75 MHz for ^13^C), a Bruker Avance III HD 400 (400 MHz for ^1^H and 101 MHz for ^13^C) or a Bruker Avance III 600 MHz (600 MHz for ^1^H and 151 MHz for ^13^C), the later equipped with a cryo-probe. All NMR spectra were acquired at 25 °C unless stated otherwise. For internal references in the ^1^H and ^13^C NMR spectra, signals of *t*-BuOH (0.05%) for D_2_O solutions (1.24 ppm/30.39(CH_3_) ppm, respectively), TMS (0.0 ppm/0.0(CH_3_) ppm, respectively) or CHCl_3_ residual peak for CDCl_3_ solutions (^1^H, 7.26 ppm) or signal of CDCl_3_ itself (^13^C, 77.16 ppm) were used. Chemical shifts *δ* are given in ppm, and coupling constants *J* are given in Hz. Additionally, s, d, t, q, p, sept, m and br express the multiplicity of signals: singlet, doublet, triplet, quartet, pentet, septet, multiplet and broad. NMR spectral characterisations of the prepared compounds are shown in ESI.

UV-Vis spectra were recorded on a Specord 50 Plus (Analytic Jena) spectrometer in a quartz-glass cell with optical path of 1 cm. The temperature was set using a cuvette holder equipped with a Peltier thermoelectric heater.

Mass spectra were recorded using a Waters ACQUITY QDa, which is part of the Waters Arc HPLC system, and are shown in ESI. Data were processed using Empower 3 software. Samples were dissolved in water, MeOH or MeCN. HPLC was conducted on the same device using a Cortecs C18 2.7 µm, 4.6 × 50 mm column. HPLC chromatograms of the ligands that were the most effective at mediating the hydrolysis of BNPP are shown in ESI.

### 4.2. Synthesis

#### 4.2.1. 1,4,7-Trimethyl-1,4,7-triazacyclononane (**L6**)

Compound **L6** was prepared using a modified version of the procedure reported in [30].

Compound **L1** (200 mg, 1.55 mmol) was dissolved in 37% aq. formaldehyde (8 mL, 108 mmol, 70 equiv.). To the solution, formic acid (2 mL, 53 mmol, 34 equiv.) was added, and the mixture was heated to reflux for 24 h. The mixture was evaporated to dryness in vacuo, and the residue was taken in 15 mL of 10% aq. NaOH. The product was extracted with CHCl_3_ (3 × 20 mL), and the organic phases were combined and dried over Na_2_SO_4_. The drying agent was filtered off, and volatiles were evaporated in vacuo, affording 232 mg (87%) of **L6** in the form of a colourless oil.

NMR (CDCl_3_): ^1^H: *δ* 2.36 (s, 9H, C*H*_3_); 2.65 (s, 12H, C*H*_2_). ^13^C{^1^H}: *δ* 46.76 (3C, *C*H_3_); 57.06 (6C, *C*H_2_).

MS-ESI (+): 172.3 ([M + H]^+^, calc. 172.2).

#### 4.2.2. 1,4,7-Tri(*iso*-propyl)-1,4,7-triazacyclononane (**L7**)

The compound **L1** (1.00 g, 7.74 mmol) was dissolved in MeCN (50 mL). Potassium carbonate (6.41 g, 46.4 mmol, 6 equiv.) and *iso*-propyl bromide (3.14 g, 25.5 mmol, 3.3 equiv.) were added, and the mixture was stirred under a condenser at 60 °C for 24 h. The solids were filtered off using a S4 frit, and the filtrate was evaporated in vacuo. The residue was dissolved in CHCl_3_ (20 mL) and extracted with water (2 × 20 mL). The organic fraction was dried over Na_2_SO_4_, filtered and evaporated, yielding **L7** as a colourless oil (1.43 g, 72%).

NMR (CDCl_3_): ^1^H *δ* 0.96 (d, ^3^*J*_HH_ = 6.5, 18H, C*H*_3_); 2.63 (s, 12H, C*H*_2_); 2.86 (sept, ^3^*J*_HH_ = 6.4, 3H, C*H*). ^13^C{^1^H}: *δ* 18.33 (6C, *C*H_3_); 52.73 (6C, *C*H_2_); 54.30 (3C, *C*H).

MS-ESI (+): 256.4 ([M + H]^+^, calc. 256.3).

#### 4.2.3. 1-Benzyl-4,7-dimethyl-1,4,7-triazacyclononane (**L8**)

Compound L8 was prepared and isolated using a procedure analogous to the procedure described above for **L6**. Starting with 3.00 g (13.7 mmol) of **L3**, 37% aq. formaldehyde (50 mL, 670 mmol, 49 equiv.) and formic acid (20 mL, 530 mmol, 39 equiv.), product **L8** was obtained in the form of a yellowish oil. Yield: 3.23 g (95%).

NMR (CDCl_3_): ^1^H: *δ* 2.35 (s, 6H, C*H*_3_); 2.66 (m, 4H, C*H*_2_(cycle)); 2.73 (m, 4H, C*H*_2_(cycle)); 2.82 (s, 4H, C*H*_2_(cycle)); 3.66 (s, 2H, C*H*_2_Ph); 7.30 (m, 5H, *arom.*). ^13^C{^1^H}: *δ* 46.61 (2C); 56.04 (2C); 56.82 (2C); 57.09 (2C); 63.42 (1C, *C*H_2_Ph); 126.75 (1C), 129.08 (2C), 129.12 (2C) and 140.25 (1C), all *arom.*

MS-ESI (+): 248.3 ([M + H]^+^, calc. 248.4).

#### 4.2.4. 1-Benzyl-4,7-di(*iso*-propyl)-1,4,7-triazacyclononane (**L9**)

Compound **L9** was prepared and isolated using a procedure analogous to the procedure described above for **L7**. **L3**∙2HCl∙H_2_O (2.00 g, 6.44 mmol) was suspended in MeCN (50 mL), and K_2_CO_3_ (2.67 g, 19.3 mmol, 3 equiv.) was added. To the stirred suspension, *iso*-propyl bromide (3.96 g, 32.2 mmol, 5 equiv.) was added, and the mixture was stirred at 60 °C overnight. After a work-up similar to **L7**, product **L9** was obtained in the form of a yellowish oil. Yield: 1.93 g (98%).

NMR (CDCl_3_): ^1^H: *δ* 0.97 (d, ^3^*J*_HH_ = 6.4, 12H, C*H*_3_); 2.50–2.72 (m, 8H, C*H*_2_(cycle)); 2.81–2.97 (m, 6H, C*H*_2_(cycle) + C*H*); 3.67 (s, 2H, C*H*_2_Ph); 7.30 (m, 5H, *arom.*). ^13^C{^1^H}: *δ* 18.42 (4C, *C*H_3_); 52.41, 52.76 and 54.79 (all 2C, *C*H_2_(cycle)); 55.37 (2C, *C*H); 62.21 (1C, *C*H_2_Ph); 126.63 (1C), 128.13 (2C), 129.02 (2C) and 140.52 (1C), all *arom*.

MS-ESI (+): 304.2 ([M + H]^+^, calc. 304.3).

#### 4.2.5. 1,4-Dimethyl-1,4,7-triazacyclononane (**L4**) and 1,4-di(*iso*-propyl)-1,4,7-triazacyclononane (**L5**)

Compound **L8** or **L9** (2.00 g) was dissolved in a mixture of methanol (30 mL), water (5 mL) and acetic acid (10 mL). A Pd/C catalyst (15% wt., 200 mg) was added, and the reaction mixture was stirred overnight at room temperature under a hydrogen atmosphere at atmospheric pressure (balloon). The suspension was filtered, and the filtrate was evaporated to dryness and than co-evaporated with water (3 × 20 mL). The residue was then taken into 20 mL of 10% aq. NaOH, and the product (**L4** or **L5**, respectively) was extracted using CHCl_3_ (3 × 30 mL). Organic fractions were combined and dried over Na_2_SO_4_. The drying agent was filtered off, and the volatiles were evaporated in vacuo, providing products in the form of light-yellow oils.

**L4**: Yield: 1.08 g (85%).

NMR (CDCl_3_): ^1^H: *δ* 2.64 (m, 4H, C*H*_2_(cycle)); 2.50 (m, 8H, C*H*_2_(cycle)); 2.37 ppm (s, 6H, C*H*_3_). ^13^C{^1^H}: *δ* 45.46, 46.22, 53.40 and 54.45 (all 2C, *C*H_2_(cycle) and *C*H_3_).

MS-ESI (+): 158.2 ([M + H]^+^, calc. 158.3).

**L5**: Yield: 1.16 g (84%).

NMR (CDCl_3_): ^1^H: *δ* 1.00 (d, ^3^*J*_HH_ = 6.7, 12H, C*H*_3_); 2.48 (s, 4H, C*H*_2_(cycle)); 2.58 (m, 4H, C*H*_2_(cycle)), 2.68 (m, 4H, C*H*_2_(cycle)); 2.86 (sept, ^3^*J*_HH_ = 6.7, 2H, C*H*). ^13^C{^1^H}: *δ* 18.81 (4C, *C*H_3_); 47.05, 47.78 and 48.99 (all 2C, *C*H_2_(cycle)); 53.05 (s, 2C, *C*H).

MS-ESI (+): 214.3 ([M + H]^+^, calc. 214.2).

#### 4.2.6. 1-[ω-(Phthalimido)-ALKYL]-4,7-di(*iso*-propyl)-1,4,7-triazacyclononane (alkyl: **2a**—propyl, **2b**—butyl, **2c**—pentyl)

General procedure: the compound **L5** was dissolved in MeCN; 30 mL per 1 g of the starting compound was used. Then, K_2_CO_3_ (3 equiv.) and alkylating agent **1a**–**c** (1.1 equiv.) were added, and the mixture was stirred at 60 °C for 2 days. Then, the mixture was filtered through an S4 frit and the filtrate was evaporated to dryness. The residue was dissolved in CHCl_3_ (30 mL) and extracted using water (2 × 20 mL). The organic fraction was dried over Na_2_SO_4_ and evaporated, yielding a bright yellow oil which was purified using flash chromatography on reverse-phase silica gel (GM1). After combining and evaporating the fractions containing the product, compounds were isolated as oily materials containing non-stoichiometrical amounts of TFA, which prevented the exact calculation of the yields (but no solvent remains were present, according to NMR spectroscopy).

**2a**: Starting from 0.45 g (2.1 mmol) of **L5**, the product **2a**·*x*TFA was obtained in a yield of 0.63 g.

NMR (D_2_O): ^1^H: *δ* 1.38 (d, ^3^*J*_HH_ = 6.5, 12H, C*H*_3_); 1.89–2.02 (m, 2H, CH_2_C*H*_2_CH_2_); 2.82–2.89 (m, 4H, C*H*_2_(cycle)); 3.03 (sept, ^3^*J*_HH_ = 6.4, 2H, C*H*); 3.10 (br s, 2H, C*H*_2_CH_2_CH_2_); 3.24 (br s, 2H, CH_2_CH_2_C*H*_2_); 3.54–3.80 (m, 8H, C*H*_2_(cycle)); 7.82 (m, 4H, *arom.*). ^13^C{^1^H}: *δ* 14.91 (4C, *C*H_3_); 17.18, 22.66 and 35.67 (all 1C, *C*H_2_*C*H_2_*C*H_2_); 45.25, 47.39, 51.65 and 59.85 (all 2C, *C*H_2_(cycle) and *C*H); 123.34, 131.21 and 134.76, (all 2C, *arom.*); 170.63 (2C, *C*O).

MS-ESI (+): 401.5 ([M + H]^+^, calc. 401.3).

**2b**: Starting from 0.48 g (2.2 mmol) of **L5**, the product **2b**·*x*TFA was obtained in a yield of 0.71 g.

NMR (D_2_O): ^1^H: *δ* 1.35 (d, ^3^*J*_HH_ = 7.2, 12H, C*H*_3_); 1.53–1.71 (m, 4H, CH_2_C*H*_2_C*H*_2_CH_2_); 2.83–2.91 (m, 2H, C*H*_2_CH_2_CH_2_CH_2_); 2.99–3.17 (m, 4H, C*H*_2_(cycle)); 3.24 (sept, ^3^*J*_HH_ = 6.9, 2H, C*H*); 3.34–3.49 (m, 2H, CH_2_CH_2_CH_2_C*H*_2_); 3.56 (s, 4H, C*H*_2_(cycle)); 3.66 (m, 4H C*H*_2_(cycle)); 7.79 (m, 4H, *arom*.). ^13^C{^1^H}: *δ* 20.19 (4C, *C*H_3_); 22.17, 26.17, 42.27 and 50.35 (all 1C, *C*H_2_*C*H_2_*C*H_2_*C*H_2_); 52.26, 53.08, 59.61 and 64.55 (all 2C, *C*H_2_(cycle) and *C*H); 128.32, 136.16 and 139.81 (all 2C, *arom.*); 175.75 (2C, *C*O).

MS-ESI (+): 415.4 ([M + H]^+^, calc. 415.3).

**2c**: Starting from 0.66 g (3.1 mmol) of **L5**, the product **2c**·*x*TFA was obtained in a yield of 0.94 g.

NMR (D_2_O): ^1^H: *δ* 1.35 (d, ^3^*J*_HH_ = 6.1, 12H, C*H*_3_); 1.63–1.71 (m, 6H, CH_2_C*H*_2_C*H*_2_C*H*_2_CH_2_); 2.83–2.95 (m, 2H, C*H*_2_CH_2_CH_2_CH_2_CH_2_); 3.07–3.29 (m, 6H, C*H*_2_(cycle) and CH_2_CH_2_CH_2_CH_2_C*H*_2_), 3.41 (sept, ^3^*J*_HH_ = 6.1, 2H, C*H*); 3.50 (s, 4H, C*H*_2_(cycle)); 3.59–3.65 (m, 4H C*H*_2_(cycle)); 7.79 (m, 4H, *arom*.). ^13^C{^1^H}: *δ* 18.59 (4C, *C*H_3_); 19.93, 25.97, 26.49, 40.46 and 48.50 (all 1C, *C*H_2_*C*H_2_*C*H_2_*C*H_2_*C*H_2_); 49.88, 51.41, 58.50 and 62.07 (all 2C, *C*H_2_(cycle) and *C*H); 126.27, 134.23 and 137.73 (all 2C, *arom.*); 173.79 (2C, *C*O).

MS-ESI (+): 429.5 ([M + H]^+^, calc. 429.3).

#### 4.2.7. 1-[ω-(Amino)-alkyl]-4,7-di(*iso*-propyl)-1,4,7-triazacyclononane (alkyl: **L10a**—propyl, **L10b**—butyl, **L10c**—pentyl)

General procedure: the chosen compound **2a**–**c** (in the form of non-stoichiometric trifluoroacetate) was dissolved in a mixture of EtOH (10 mL) and hydrazine monohydrate (80%, 10 mL), and the reaction mixture was refluxed overnight. The solvents were evaporated, the solid residue was dissolved in EtOH and filtered to remove most of phtalhydrazide formed during deprotection and the filtrate was evaporated to dryness. However, there was still some phtalhydrazide present in the residue; therefore, it was further hydrolysed. The residue was dissolved in EtOH (20 mL), and an excess of solid NaOH (1 g) was added. The mixture was heated to reflux overnight. Volatiles were evaporated, and the residue was dissolved in water and extracted using DCM (3 × 30 mL). Organic fractions were combined, dried over Na_2_SO_4_, filtered and evaporated, yielding **L10a**–**c**, respectively, as viscous, light-yellow oils.

**L10a**: Starting from 0.63 g of **2a**∙*x*TFA, product **L10a** was obtained in a yield of 0.37 g (65% to **L5** over two steps).

NMR (CDCl_3_): ^1^H: *δ* 0.97 (d, ^3^*J*_HH_ = 6.6, 12H, C*H*_3_); 1.63 (p, ^3^*J*_HH_ = 6.8, 2H, CH_2_C*H*_2_CH_2_); 2.56–2.61 (m, 6H, C*H*_2_(cycle) and C*H*_2_CH_2_CH_2_); 2.65–2.72 (m, 4H, C*H*_2_(cycle)); 2.72–2.79 (m, 2H, CH_2_CH_2_C*H*_2_); 2.81–2.86 (m, 4H, C*H*_2_(cycle)); 2.91 (sept, ^3^*J*_HH_ = 6.5, 2H, C*H*). ^13^C{^1^H}: *δ* 18.31 (4C, *C*H_3_); 31.69 (2C, CH_2_*C*H_2_CH_2_); 40.80, 52.39, 52.47, 54.70, 55.38 and 55.97 (*C*H_2_CH_2_*C*H_2_, *C*H_2_(cycle) and *C*H).

MS-ESI (+): 271.4 ([M + H]^+^, calc. 271.3).

**L10b**: Starting from 0.71 g of **2b**∙*x*TFA, product **L10b** was obtained in a yield of 0.44 g (69% to **L5** over two steps).

NMR (CDCl_3_): ^1^H: *δ* 1.00 (d, ^3^*J*_HH_ = 6.7, 12H, C*H*_3_); 1.34–1.50 (m, 2H, CH_2_C*H*_2_CH_2_CH_2_); 1.50–1.65 (m, 2H, CH_2_CH_2_C*H*_2_CH_2_); 2.62 (t, ^3^*J*_HH_ = 7.1, 2H, C*H*_2_CH_2_CH_2_CH_2_); 2.69–2.80 (m, 6H, C*H*_2_(cycle) and CH_2_CH_2_CH_2_C*H*_2_); 2.80–2.87 (m, 4H, C*H*_2_(cycle)); 2.89 (s, 4H, C*H*_2_(cycle)); 3.24 (sept, ^3^*J*_HH_ = 6.6, 2H, C*H*). ^13^C{^1^H}: *δ* 18.41 (4C, *C*H_3_); 25.04 and 31.57 (all 1C, CH_2_*C*H_2_*C*H_2_CH_2_); 42.07, 51.89, 52.04, 54.64, 54.79 and 57.83 (*C*H_2_CH_2_CH_2_*C*H_2_, *C*H_2_(cycle) and *C*H).

MS-ESI (+): 285.2 ([M + H]^+^, calc. 285.3).

**L10c**: Starting from 0.94 g of **2c**∙*x*TFA, product **L10c** was obtained in a yield of 0.63 g (68% to **L5** over two steps).

NMR (CDCl_3_): ^1^H: *δ* 0.99 (d, ^3^*J*_HH_ = 6.7, 12H, C*H*_3_); 1.32 (p, ^3^*J*_HH_ = 7.7, 2H, CH_2_CH_2_C*H*_2_CH_2_CH_2_); 1.43–1.53 (m, 4H, CH_2_C*H*_2_CH_2_C*H*_2_CH_2_); 2.46–2.56 (m, 2H, C*H*_2_CH_2_CH_2_CH_2_C*H*_2_); 2.56–2.64 (m, 4H, C*H*_2_(cycle)); 2.64–2.76 (m, 6H, C*H*_2_(cycle) and C*H*_2_CH_2_CH_2_CH_2_C*H*_2_); 2.83–2.90 (m, 4H, C*H*_2_(cycle)); 2.93 (sept, ^3^*J*_HH_ = 6.6, 2H, C*H*). ^13^C{^1^H}: *δ* 18.30 (4C, *C*H_3_); 24.16, 25.98, 29.85 and 30.40 (all 1C, CH_2_*C*H_2_CH_2_*C*H_2_CH_2_); 40.78, 46.80, 49.82, 53.80 and 55.45 (*C*H_2_CH_2_CH_2_CH_2_*C*H_2_, *C*H_2_(cycle) and *C*H).

MS-ESI (+): 299.4 ([M + H]^+^, calc. 299.3).

#### 4.2.8. 1-[ω-(Thiazole-4-ylcarboxamido)alkyl]-4,7-di(*iso*-propyl)-1,4,7-triazacyclononane (alkyl: **L11a**—propyl, **L11b**—butyl, **L11c**—pentyl)

General procedure: the chosen amine **L10a**–**c** (1 equiv.) was dissolved in DMF (20 mL). To this solution, Et_3_N (1 equiv.) was added. In another flask, to a prepared solution of thiazole-4-carboxylic acid (1.5 equiv.) and Et_3_N (2 equiv.) in DMF (20 mL), HATU (1.5 equiv.) was added. After 5 min, the solution containing amine **L10a**–**c** was added dropwise to the flask containing the thiazole-4-carboxylic acid–HATU solution, and the resulting mixture was stirred at room temperature overnight. The reaction mixture was evaporated to dryness in vacuo, and the residue was purified using column chromatography on silica gel (EtOH:conc. aq. NH_3_ 5:1). Product-containing fractions were combined and evaporated, yielding the ligands **L11a**–**c** as light yellow oils.

**L11a**: Starting from 0.3 g (1.11 mmol) of **L10a**, product **L11a** was obtained in a yield of 0.05 g (12%).

NMR (CDCl_3_): ^1^H: *δ* 1.03 (d, ^3^*J*_HH_ = 6.6, 12H, C*H*_3_); 1.83–1.95 (m, 2H, CH_2_C*H*_2_CH_2_); 2.67 (br s, 4H, C*H*_2_(cycle)); 2.82 (br s, 6H C*H*_2_(cycle) and C*H*_2_CH_2_CH_2_); 3.00 (sept, ^3^*J*_HH_ = 6.7, 2H, C*H*); 3.09 (br s, 4H, C*H*_2_(cycle)); 3.56 (q, ^3^*J*_HH_ = 6.4, 2H, C*H*_2_NHCO); 8.26 (d, ^4^*J*_HH_ = 2.1, 1H, CC*H*S); 8.29 (s, 1H, N*H*); 8.76 (d, ^4^*J*_HH_ = 2.1, 1H, NC*H*S). ^13^C{^1^H}: *δ* 18.25 (4C, *C*H_3_); 18.66, 26.49 and 37.74 (all 1C, *C*H_2_*C*H_2_*C*H_2_); 50.18, 52.53 and 54.20 (very br, *C*H_2_(cycle)); 54.63 (2C, *C*H); 123.05, 151.40 and 152.66 (all 1C, *thiazole*); 161.32 (1C, *C*O).

MS-ESI (+): 382.4 ([M + H]^+^, calc. 382.3).

**L11b**: Starting from 0.13 g (0.5 mmol) of **L10b**, product **L11b** was obtained in a yield of 0.03 g (17%).

NMR: ^1^H (CD_3_OD): *δ* 1.03 (d, ^3^*J*_HH_ = 6.6, 12H, C*H*_3_); 1.50–1.74 (m, 4H, CH_2_C*H*_2_C*H*_2_CH_2_); 2.54–2.64 (m, 2H, C*H*_2_CH_2_CH_2_CH_2_); 2.68 (s, 4H, C*H*_2_(cycle)); 2.71-2.80 (m, 4H, C*H*_2_(cycle)); 3.00 (sept, ^3^*J*_HH_ = 6.8, 2H, C*H*); 3.44 (t, *J*_HH_ = 6.3, 2H, C*H*_2_NHCO); 8.23 (d, ^4^*J*_HH_ = 2.0, 1H, CC*H*S); 9.00 (d, ^4^*J*_HH_ = 2.0, 1H, NC*H*S). ^13^C{^1^H} (CDCl_3_): *δ* 18.32 (4C, *C*H_3_); 23.73 and 27.12 (all 1C, CH_2_*C*H_2_*C*H_2_CH_2_); 38.77, 46.36, 46.71, 49.62, 53.69 and 55.17 (*C*H_2_CH_2_CH_2_*C*H_2_, *C*H_2_(cycle) and *C*H); 123.23, 151.41 and 153.14 (all 1C, *thiazole*); 161.37 (1C, *C*O).

MS-ESI (+): 396.4 ([M + H]^+^, calc. 396.3).

**L11c**: Starting from 0.30 g (1.01 mmol) of **L10c**, product **L11c** was obtained in a yield of 0.12 g (29%).

NMR (CD_3_OD): ^1^H: *δ* 1.02 (d, ^3^*J*_HH_ = 6.5, 12H, C*H*_3_); 1.36–1.46 (m, 2H, CH_2_CH_2_C*H*_2_CH_2_CH_2_); 1.50–1.59 (m, 2H, CH_2_C*H*_2_CH_2_CH_2_CH_2_); 1.66 (p, ^3^*J*_HH_ = 7.2, 2H, CH_2_CH_2_CH_2_C*H*_2_CH_2_); 2.55 (t, ^3^*J*_HH_ = 7.6, 2H, C*H*_2_CH_2_CH_2_CH_2_CH_2_); 2.64 (s, 4H, C*H*_2_(cycle)), 2.69–2.77 (m, 4H, C*H*_2_(cycle)), 2.89–2.99 (m, 6H, C*H*_2_(cycle) and C*H*); 3.41 (t, ^3^*J*_HH_ = 7.0, 2H, C*H*_2_NHCO); 8.22 (d, ^4^*J*_HH_ = 2.0, 1H, CC*H*S); 8.99 (d, ^4^*J*_HH_ = 2.2, 1H, NC*H*S). ^13^C{^1^H}: 18.48 (4C, *C*H_3_); 25.89, 28.01, 30.43 and 40.23 (all 1C, CH_2_*C*H_2_*C*H_2_*C*H_2_CH_2_); 50.35, 50.80 and 52.95 (all very br s, *C*H_2_(cycle)), 55.43 and 57.81 (*C*H_2_CH_2_CH_2_CH_2_*C*H_2_ and *C*H); 124.61, 152.07 and 155.36 (all 1C, *thiazole*); 163.44 (1C, *C*O).

MS-ESI (+): 410.4 ([M + H]^+^, calc. 410.3).

#### 4.2.9. *N*-(2-Aminoethyl)-4-thiazolecarboxamide (**5**)

To a stirring solution of methyl-4-thiazolecarboxylate **4** (0.8 g, 5.6 mmol) in MeOH (30 mL), ethylenediamine (1 equiv., 0.34 g) was added, and the mixture was refluxed overnight. The solvent was evaporated, and the crude product containing both the monomeric product and its dimer was purified via column chromatography on silica gel, using EtOH/NH_3_ 20/1 (*v*/*v*) as a mobile phase. Fractions containing the product were combined and evaporated, yielding 0.32 g (34%) of **5** in the form of a yellow oil.

NMR (CDCl_3_): ^1^H: *δ* 1.69 (s, 2H, N*H*_2_); 2.93 (t, ^3^*J*_HH_ = 6.0, 2H, C*H*_2_), 3.50 (q, ^3^*J*_HH_ = 6.0, 2H, C*H*_2_), 7.77 (s, 1H, N*H*), 8.16 (d, ^4^*J*_HH_ = 2.1, 1H, CC*H*S), 8.74 (d, ^4^*J*_HH_ = 2.2, 1H, NC*H*S). ^13^C{^1^H}: *δ* 41.63 and 42.24 (all 1C, *C*H_2_*C*H_2_); 123.24, 151.23 and 152.83 (all 1C, *thiazole*); 161.40 (1C, *C*O).

MS-ESI: (+) 144.1 ([M + H]^+^, calc. 144.0).

#### 4.2.10. *N*-[2-(2-Chloroacetyl)amidoethyl]-4-thiazolecarboxamide (**6a**) and N-[2-(3-chloropropionyl)amidoethyl]-4-thiazolecarboxamide (**6b**)

To a stirring solution of *N*-(2-aminoethyl)-4-thiazolecarboxamide **5** and Et_3_N (1.2 equiv.) in dry DCM, chloroacetyl chloride (1.2 equiv.) in dry DCM was added dropwise. During the addition, the mixture was cooled to 0 °C in an ice–water bath. After the addition, the mixture was allowed to warm up to room temperature and was left stirring overnight. Volatiles were evaporated, and the resulting reaction mixture was purified using column chromatography on silica gel, using MeOH:conc. aq. NH_3_ 20:1 (**6a**) or ethyl acetate/methanol 3:1 (**6b**) as mobile phases. Fractions containing products were combined and evaporated, yielding compound **6a** in form of a yellow powder and compound **6b** as a yellow oil.

**6a**: Starting from 0.24 g of **5**, product **6a** was obtained in a yield of 0.13 g (37%).

NMR (CDCl_3_): ^1^H: *δ* 3.57 (m, 2H, C*H*_2_CH_2_); 3.66 (m, 2H, CH_2_C*H*_2_); 4.05 (s, 2H, C*H*_2_Cl); 7.31 (s, 1H, N*H*); 7.79 (s, 1H, N*H*); 8.21 (d, ^4^*J*_HH_ = 2.1, 1H, CC*H*S), 8.78 (d, ^4^*J*_HH_ = 2.1, 1H, NC*H*S); ^13^C{^1^H}: *δ* 39.25, 40.91 and 42.65 (all 1C, *C*H_2_); 123.67, 150.52 and 152.96 (all 1C, *thiazole*); 162.10 and 166.93 (all 1C, *C*O).

MS-ESI: (+) 247.7 ([M + H]^+^, calc. 248.0).

**6b**: Starting from 0.19 g of **5**, product **6a** was obtained in a yield of 0.18 g (62%) at 70% purity, according to ^1^H NMR.

NMR (CDCl_3_): ^1^H *δ* 2.66 (t, ^3^*J*_HH_ = 6.6, 2H, C*H*_2_CO); 3.51–3.75 (m, 4H, NC*H*_2_C*H*_2_N); 3.80 (t, ^3^*J*_HH_ = 6.5, 2H, C*H*_2_Cl); 6.55 (s, 1H, N*H*); 7.80 (s, 1H, N*H*); 8.20 (d, ^4^*J*_HH_ = 2.1, CC*H*S), 8.79 (d, ^4^*J*_HH_ = 2.1, 1H, NC*H*S). ^13^C{^1^H}: *δ* 39.37, 39.71, 40.22 and 40.98 (all 1C, *C*H_2_); 123.66, 150.73 and 153.05 (all 1C, *thiazole*); 162.52 and 170.18 (all 1C, *C*O).

MS-ESI: (+) 261.7 ([M + H]^+^, calc. 262.0).

#### 4.2.11. {*N*-[*N*-(Thiazol-4-yl-carbonyl)-2-amidoethyl]amidocarbonylmethyl}-4,7-di(*iso*-propyl)-1,4,7-triazacyclononane (**L12a**) and (2-{*N*-[*N*-(thiazol-4-yl-carbonyl)-2-amidoethyl]amidocarbonyl}ethyl)-4,7-di(*iso*-propyl)-1,4,7-triazacyclononane (**L12b**)

To a stirred suspension of **L5** (ca 0.1 g, 1 equiv.) and K_2_CO_3_ (4 equiv.) in MeCN (30 mL), alkylating agent **6a** or **6b** (1.1 equiv.) was added, and the mixture was heated to reflux overnight (the alkylating reagent **6** dissolved at a higher temperature). After filtration through S4 frit, the filtrate was evaporated to dryness. Work-up for **L12a**: The residue was dissolved in DCM and extracted with water (3 × 15 mL). The organic fraction was dried over Na_2_SO_4_, filtered and evaporated. Product **L12a** was obtained in the form of light brown oil. Work-up for **L12b**: The reaction mixture was purified using reversed-phase flash chromatography (GM2) and the fraction containing the product was evaporated, yielding **L12b**·*x*TFA in the form of a light yellow oil.

**L12a**: Starting from 0.1 g (0.5 mmol) of **L5**, product **L12a** was obtained in a yield of 0.09 g (47%).

NMR (CDCl_3_): ^1^H: *δ* 0.92 (d, ^3^*J*_HH_ = 6.6, 12H, C*H*_3_); 2.53–2.64 (m, 12H, CH_2_(cycle)); 2.76–2.85 (sept, ^3^*J*_HH_ = 6.5, 2H, C*H*); 3.26 (s, 2H, NC*H*_2_CO); 3.46–3.53 (m, 2H, CONHC*H*_2_); 3.59 (q, *J*_HH_ = 5.9, 2H, CONHC*H*_2_); 7.99 (s, 1H, N*H*), 8.14 (d, ^4^*J*_HH_ = 2.1, 1H, CC*H*S); 8.75 (d, ^4^*J*_HH_ = 2.2, 1H, NC*H*S); 9.71 (s, 1H, N*H*). ^13^C{^1^H}: *δ* 18.33 (4C, *C*H_3_); 38.70, 40.27, 49.96 and 54.90 (all 2C, *C*H_2_(cycle) and *C*H); 55.36 (1C, *C*H_2_CO); 59.48 and 61.59 (all 1C, CONHC*H*_2_*C*H_2_NHCO); 122.99, 151.38 and 152.77 (all 1C, *thiazole*); 161.37 and 174.49 (all 1C, *C*O).

MS-ESI: (+) 425.4 ([M + H]^+^, calc.425.3).

**L12b**: Starting from 0.13 g (0.6 mmol) of **L5**, product **L12b**·*x*TFA was obtained in a yield of 0.22 g.

NMR (D_2_O): ^1^H (80 °C): *δ* 1.87 (d, ^3^*J*_HH_ = 6.6, 12H, C*H*_3_); 3.03 (t, ^3^*J*_HH_ = 6.9, 2H, NC*H*_2_CH_2_CO); 3.50 (t, ^3^*J*_HH_ = 6.9, 2H, NCH_2_C*H*_2_CO); 3.57 (br s, 4H, C*H*_2_(cycle)); 3.78–3.95 (br m, 4H, C*H*_2_(cycle)); 4.05–4.09 (m, 2H, NHC*H*_2_C*H*_2_NH); 4.09–4.13 (m, 2H, NHC*H*_2_C*H*_2_NH); 4.15–4.24 (m, 4H, C*H*_2_(cycle) and C*H*); 8.82 (d, ^4^*J*_HH_ = 2.0, 1H, CC*H*S); 9.59 (d, ^4^*J*_HH_ = 2.0, 1H, NC*H*S). ^13^C{^1^H} (25 °C): *δ* 15.63 (4C, *C*H_3_); 16.57, 31.28, 38.81, 38.82, 46.48 and 46.87 (*C*H_2_(cycle), *C*H and N*C*H_2_*C*H_2_CO); 49.35 and 60.00 (all 1C, CONHC*H*_2_*C*H_2_NHCO); 125.09, 148.87 and 155.91 (all 1C, *thiazole*); 163.45 and 176.26 (all 1C, *C*O).

MS-ESI: (+) 439.5 ([M + H]^+^, calc. 439.3).

#### 4.2.12. 1-[ω-(Thiophene-3-ylcarboxamido)alkyl]-4,7-di(*iso*-propyl)-1,4,7-triazacyclononane (Alkyl: **L13a**—propyl, **L13b**—butyl, **L13c**—pentyl)

General procedure: 3-thiophenecarboxylic acid was dissolved in SOCl_2_ (excess), and the mixture was heated to reflux for 3 h. Excess SOCl_2_ was evaporated, and the 3-thiophenecarbonyl chloride (**7**) formed was used in the next step without purification. To a mixture of **10a–c** (1 equiv.) and Et_3_N (3 equiv.) in DCM, 3-thiophenecarbonyl chloride (**7**) (1.1 equiv.) was added, and the mixture was left stirring at room temperature overnight. Then, the solvent was evaporated, and the product was purified using reversed-phase flash chromatography (GM2). Product-containing fractions were combined and evaporated, yielding **L13a**–**c**·*x*TFA as light-yellow oils.

**L13a**: Starting from 82 mg (0.3 mmol) of **L10a**, 24 mg of product **L13a** was obtained.

NMR (CDCl_3_): ^1^H: *δ* 1.35 (d, ^3^*J*_HH_ = 6.5, 12H, C*H*_3_); 1.89–1.99 (m, 2H, CH_2_C*H*_2_CH_2_); 2.94 (m, 2H, C*H*_2_CH_2_CH_2_); 3.01–3.52 (br m, 14H, C*H*_2_(cycle) and CH_2_CH_2_C*H*_2_); 3.60 (p, ^3^*J*_HH_ = 6.6, 2H, C*H*); 7.28 (s, 1H, N*H*); 7.32 (dd, ^3^*J*_HH_ = 5.1, ^4^*J*_HH_ = 2.9, 1H, SC*H*CH); 7.51 (dd, ^3^*J*_HH_ = 5.1, ^4^*J*_HH_ = 1.2, 1H, SCHC*H*); 8.06 (dd, ^4^*J*_HH_ = 2.9 and 1.2, 1H, SC*H*C). ^13^C{^1^H}: *δ* 16.77 (4C, *C*H_3_); 17.01 (very br, 1C, CH_2_*C*H_2_CH_2_); 25.08 (1C, *C*H_2_CH_2_CH_2_); 37.43, 46.53, 47.12, 49.09 and 59.45 (all very br, CH_2_CH_2_*C*H_2_, *C*H_2_(cycle) and *C*H); 126.47, 129.19, 136.67 and 160.84 (all 1C, *thiophene*); and 164.22 (1C, *C*O).

MS-ESI: (+) 381.5 ([**L13a** + H]^+^, calc. 381.3).

**L13b**: Starting from 57 mg (0.2 mmol) of **L10b**, 21 mg of product **L13b**·*x*TFA was obtained.

NMR (D_2_O): ^1^H: *δ* 1.33 (d, ^3^*J*_HH_ = 6.6, 12H, C*H*_3_); 1.58–1.70 (m, 4H, CH_2_C*H*_2_C*H*_2_CH_2_); 2.92–2.86 (m, 2H, CH_2_CH_2_CH_2_C*H*_2_); 3.03–3.29 (br m, 6H, C*H*_2_(cycle) and C*H*_2_CH_2_CH_2_CH_2_); 3.34–3.45 (m, 4H, C*H*_2_(cycle)); 3.53 (s, 4H, C*H*_2_(cycle)); 3.64 (sept, ^3^*J*_HH_ = 6.6, 2H, C*H*); 7.43 (dd, ^3^*J*_HH_ = 5.1, ^4^*J*_HH_ = 1.4, 1H, SCHC*H*); 7.53 (dd, ^3^*J*_HH_ = 5.1, ^4^*J*_HH_ = 2.9, 1H, SC*H*CH); 8.02 (dd, ^4^*J*_HH_ = 3.0 and 1.3, 1H, SC*H*C). ^13^C{^1^H}: *δ* 19.46 (4C, *C*H_3_); 21.29 and 25.39 (all 1C, CH_2_*C*H_2_*C*H_2_CH_2_); 43.13, 49.56, 52.43, 59.27 and 63.63 (*C*H_2_CH_2_CH_2_*C*H_2_, *C*H_2_(cycle) and *C*H); 130.02, 131.72, 133.76 and 140.11 (all 1C, *thiophene*); 170.22 (1C, *C*O).

MS-ESI: (+) 395.5 ([**L13b** + H]^+^, calc. 395.3).

**L13c**: Starting from 130 mg (0.4 mmol) of **L10c**, 85 mg of product **L13c** was obtained.

NMR (D_2_O): ^1^H: *δ* 1.30 (d, ^3^*J*_HH_ = 6.6, 12H, C*H*_3_); 1.35–1.41 (m, 2H, CH_2_CH_2_C*H*_2_CH_2_CH_2_); 1.56–1.72 (m, 4H, CH_2_C*H*_2_CH_2_C*H*_2_CH_2_); 2.95–3.46 (br m, 14H, C*H*_2_(cycle) + C*H*); 3.47–3.58 (m, 4H, C*H*_2_CH_2_CH_2_CH_2_C*H*_2_); 6.74 (s, 1H, N*H*); 7.32 (d, ^3^*J*_HH_ = 5.1, 1H, SC*H*CH); 7.46 (d, ^3^*J*_HH_ = 5.1, 1H, SCHC*H*); 7.97 (br s, 1H, SC*H*C). ^13^C{^1^H}: *δ* 16.38 and 17.49 (4C, *C*H_3_); 23.88, 24.34 and 28.77 (all 1C, CH_2_*C*H_2_*C*H_2_*C*H_2_CH_2_); 40.00, 46.28, 47.53, 49.23, 56.42 and 59.69 (*C*H_2_CH_2_CH_2_CH_2_*C*H_2_, *C*H_2_(cycle) and *C*H); 126.52, 128.08, 130.12 and 136.65 (all 1C, *thiophene*); 166.57 (1C, *C*O).

MS-ESI: (+) 409.5 ([**L13c** + H]^+^, calc. 409.3).

#### 4.2.13. 1,4-bis(*tert*-Butyloxycarbonyl)-1,4,7-triazonane (**8**)

Compound **8** was prepared using a modified procedure previously published in [24]. To a stirred solution of TACN (1.50 g, 11.6 mmol) and Et_3_N (2.27 mL, 16.3 mmol, 1.4 equiv.) in anhydrous CHCl_3_ (20 mL) in an ice bath, Boc_2_O (4.31 g, 19.7 mmol, 1.7 equiv.) dissolved in anhydrous CHCl_3_ (10 mL) was added dropwise over the course of 1 h. The mixture was then slowly warmed to room temperature and left stirring overnight. The solvent was removed under reduced pressure to yield a viscid, colourless oil which was purified via silica gel chromatography (DCM:MeOH 15:1) to produce pure product **8** as a colourless oil (2.12 g, 67%). The characterisation data are identical to those found in [24].

#### 4.2.14. 3-(3-Bromopropyl)thiophene (**9a**) and 3-(5-bromopentyl)thiophene (**9c**)

General procedure: compounds **9a** and **9c** were prepared from 3-bromothiophene in a procedure analogous to the procedure described for compound **9b** [27], using corresponding di-bromoalkyl reagents. The reaction mixtures were purified via column chromatography using hexane:EtOAc 100:3 as a mobile phase (TLC visualisation: vaniline), affording **9a**–**c** in the form of colourless liquids.

**9a** Starting from 6.0 g (37 mmol) of 3-bromothiophene, 2.2 g of **9a** was obtained after purification (27%).

NMR (CDCl_3_): ^1^H: *δ* 2.10–2.18 (m, 2H, CH_2_C*H*_2_CH_2_); 2.75–2.83 (m, 2H, C*H*_2_CH_2_CH_2_); 3.38 (t, *J*_HH_ = 6.6, 2H, CH_2_CH_2_C*H*_2_); 6.93 (dd, ^3^*J*_HH_ = 5.1, ^4^*J*_HH_ = 1.3, 1H, SCHC*H*); 6.97 (dd, ^4^*J*_HH_ = 3.1 and 1.3, 1H, SC*H*C); 7.25 (dd, ^3^*J*_HH_ = 5.1, ^4^*J*_HH_ = 3.1, SC*H*CH). ^13^C{^1^H}: *δ* 28.49, 33.24 and 33.38 (all 1C, *C*H_2_*C*H_2_*C*H_2_); 120.90, 125.75, 128.15 and 140.77 (all 1C, *thiophene*).

**9b**: Starting from 6.0 g (37 mmol) of 3-bromothiophene, 3.5 g of **9b** was obtained after purification (44%). Characterisation data are identical with literature [27].

**9c**: Starting from 6.0 g (37 mmol) of 3-bromothiophene, 3.9 g of **9c** was obtained after purification (45%).

NMR (CDCl_3_): ^1^H: *δ* 1.42–1.54 (m, 2H, CH_2_CH_2_C*H*_2_CH_2_CH_2_); 1.59–1.70 (m, 2H, CH_2_C*H*_2_CH_2_CH_2_CH_2_); 1.82–1.94 (m, 2H, CH_2_CH_2_CH_2_C*H*_2_CH_2_); 2.60–2.68 (m, 2H, C*H*_2_CH_2_CH_2_CH_2_CH_2_); 3.40 (t, ^3^*J*_HH_ = 6.8, 2H, CH_2_CH_2_CH_2_CH_2_C*H*_2_); 6.94–6.90 (m, 2H, SCHC*H* and SC*H*C); 7.24 (dd, ^3^*J*_HH_ = 4.7, ^4^*J*_HH_ = 3.1, 1H, SC*H*CH). ^13^C{^1^H}: *δ* 27.94, 29.81, 30.16, 32.74 and 33.89 (all 1C, *C*H_2_*C*H_2_*C*H_2_*C*H_2_*C*H_2_); 120.13, 125.38, 128.27 and 142.71 (all 1C, *thiophene*).

#### 4.2.15. 1-[ω-(Thiophene-3-yl)-alkyl]-4,7-bis(*tert*-butyloxycarbonyl)-1,4,7-triazacyclononane (alkyl: **10a**—propyl, **10b**—butyl, **10c**—pentyl)

General procedure: To a suspension of di-protected macrocycle **8** (ca 1 g) and K_2_CO_3_ (3 equiv.) in MeCN (30 mL), alkylating agent **9a**–**c** (1.1 equiv.) was added. The mixture was heated to 60 °C and left stirring overnight. Then the mixture was filtered through an S4 frit and evaporated. The residue was purified via chromatography, using: **10a**,**b** on silica gel and hexane/ethylacetate 4.5:1 as a mobile phase. Product-containing fractions were combined and evaporated, yielding products **10a**,**b** as colourless oils. **10c** was purified via flash chromatography (GM1), yielding **10c** as a yellow oil in trifluoroacetate form.

**10a**: Starting from 0.75 g (2.3 mmol) of **8**, product **10a** was obtained after purification in a yield of 0.75 g (73%).

NMR (CDCl_3_): ^1^H: *δ* 1.45 (s, 18H, C*H*_3_); 1.69–1.87 (m, 2H, CH_2_C*H*_2_CH_2_); 2.41–2.77 (m, 8H, C*H*_2_CH_2_C*H*_2_ and C*H*_2_(cycle)); 3.14–3.33 (m, 4H, C*H*_2_(cycle)); 3.37–3.56 (m, 4H, C*H*_2_(cycle)); 6.93–6.99 (m, 2H, SCHC*H* and SC*H*C); 7.18–7.27 (m, 1H, SC*H*CH). ^13^C{^1^H}: *δ* 28.67 (6C, *C*H_3_); 49.63, 50.62, 53.65, 54.00, 56.32 and 56.44 (*C*H_2_*C*H_2_*C*H_2_ and *C*H_2_(cycle)); 79.43 (2C, *C*(CH_3_)_3_; 119.99, 125.25, 128.21 and 142.11 (all 1C, *thiophene*); 155.41 (2C, *C*O).

MS-ESI: (+) 454.5 ([M + H]^+^, calc. 454.3).

**10b**: Starting from 1.28 g (3.9 mmol) of **8**, product **10b** was obtained after purification in a yield of 0.98 g (54%).

NMR (CDCl_3_): ^1^H: *δ* 1.48 (s, 18H, C*H*_3_); 1.55–1.70 (m, 4H, CH_2_C*H*_2_C*H*_2_CH_2_); 2.35–2.76 (br m, 8H, C*H*_2_CH_2_CH_2_C*H*_2_ and C*H*_2_(cycle)); 3.12–3.32 (m, 4H, C*H*_2_(cycle)); 3.38–3.52 (br m, 4H, C*H*_2_(cycle)); 6.93–6.99 (m, 2H, SCHC*H* and SC*H*C); 7.21–7.25 (m, 1H, SC*H*CH). ^13^C{^1^H}: *δ* 28.49 (6C, *C*H_3_); 28.68 and 30.31 (all 1C, CH_2_*C*H_2_*C*H_2_CH_2_); 49.92, 50.77, 51.00, 54.11, 56.79 (*C*H_2_CH_2_CH_2_*C*H_2_ and *C*H_2_(cycle)); 79.54 (2C, *C*(CH_3_)_3_; 120.11, 125.32, 128.32 and 141.92 (all 1C, *thiophene*); 155.69 (2C, *C*O).

MS-ESI: (+) 468.5 ([M + H]^+^, calc. 468.3).

**10c**: Starting from 0.45 g (1.4 mmol) of **8**, product **10c**·*x*TFA was obtained after purification in a yield of 0.4 g.

NMR (CD_3_OD): ^1^H: *δ* 1.51 (s, 18H, C*H*_3_); 1.65–1.85 (m, 4H, CH_2_C*H*_2_C*H*_2_CH_2_CH_2_); 2.68 (t, *J*_HH_ = 7.5, 2H, CH_2_CH_2_CH_2_C*H*_2_CH_2_); 3.32–3.83 (br m, 16H, C*H*_2_CH_2_CH_2_CH_2_C*H*_2_ and C*H*_2_(cycle)); 6.96 (dd, ^3^*J*_HH_ = 5.0, ^4^*J*_HH_ = 1.3, 1H, SCHC*H*); 7.05 (dd, ^4^*J*_HH_ = 2.9 and 1.3, 1H, SC*H*C); 7.32 (dd, ^3^*J*_HH_ = 5.0, ^4^*J*_HH_ = 2.9, 1H, SC*H*CH). ^13^C{^1^H}: *δ* 25.15 and 27.06 (all 1C, CH_2_*C*H_2_*C*H_2_CH_2_CH_2_) 28.60 (6C, *C*H_3_); 30.71, 31.01, 45.16, 52.11, 52.81 and 57.56 (*C*H_2_CH_2_CH_2_*C*H_2_*C*H_2_ and *C*H_2_(cycle)); 82.92 (2C, *C*(CH_3_)_3_; 121.20, 126.32, 129.09 and 143.54 (all 1C, *thiophene*); 157.24 (2C, *C*O).

MS-ESI: (+) 482.5 ([M + H]^+^, calc. 482.3).

#### 4.2.16. 1-[ω-(Thiophene-3-yl)-alkyl]-1,4,7-triazacyclononane (alkyl: **11a**—propyl, **11b**—butyl, **11c**—pentyl)

General procedure: compound **10a**–**c**·*x*TFA was suffused with 6M HCl and left stirring at room temperature overnight. During this time, the originally insoluble oil dissolved as product **11a**–**c** was formed. The solution was evaporated to dryness, yielding **11a**–**c**·*x*HCl as a white or slightly yellow powder.

**11a**: Starting from 0.75 g (1.7 mmol) of **10a**, product **11a**·*x*HCl was obtained in a yield of 0.55 g.

NMR (D_2_O): ^1^H: *δ* 1.90–2.00 (m, 2H, CH_2_C*H*_2_CH_2_); 2.69 (t, ^3^*J*_HH_ = 7.4, 2H, C*H*_2_CH_2_CH_2_); 2.88–2.95 (m, 2H, CH_2_CH_2_C*H*_2_); 3.17 (dd, ^3^*J*_HH_ = 6.8 and 4.9, 4H, C*H*_2_(cycle)); 3.34 (dd, ^3^*J*_HH_ = 6.8 and 4.9, 4H, C*H*_2_(cycle)); 3.51 (s, 4H, C*H*_2_(cycle)); 7.08 (dd, ^3^*J*_HH_ = 5.0, ^4^*J*_HH_ = 1.3, 1H, SCHC*H*); 7.15 (dd, ^4^*J*_HH_ = 2.9 and 1.3, 1H, SC*H*C); 7.43 (dd, ^3^*J*_HH_ = 5.0, ^4^*J*_HH_ = 2.9, 1H, SC*H*CH). ^13^C{^1^H}: *δ* 25.48 and 27.52 (all 1C, CH_2_*C*H_2_*C*H_2_); 42.55, 43.35, 48.58 and 55.62 (all 2C, *C*H_2_(cycle) and *C*H_2_CH_2_CH_2_); 121.62, 126.95, 129.00 and 142.56 (all 1C, *thiophene*).

MS-ESI: (+) 254.3 ([M + H]^+^, calc. 254.2).

**11b**: Starting from 0.36 g (0.8 mmol) of **10b**, product **11b**·*x*HCl was obtained in a yield of 0.26 g.

NMR (D_2_O): ^1^H: *δ* 1.64 (m, 4H, CH_2_C*H*_2_C*H*_2_CH_2_); 2.70 (t, ^3^*J*_HH_ = 6.7, 2H, C*H*_2_CH_2_CH_2_CH_2_); 2.94 (t, ^3^*J*_HH_ = 7.6, 2H, CH_2_CH_2_CH_2_C*H*_2_); 3.16 (t, ^3^*J*_HH_ = 5.9, 4H, C*H*_2_(cycle)); 3.30 (t, ^3^*J*_HH_ = 5.8, 4H, C*H*_2_(cycle)); 3.44 (s, 4H, C*H*_2_(cycle)); 7.07 (dd, ^3^*J*_HH_ = 5.0, ^4^*J*_HH_ = 1.3, 1H, SCHC*H*); 7.13 (dd, ^4^*J*_HH_ = 3.2 and 1.3, 1H, SC*H*C); 7.42 (dd, ^3^*J*_HH_ = 5.0, ^4^*J*_HH_ = 3.2, 1H, SC*H*CH). ^13^C{^1^H}: *δ* 23.97, 27.83 and 29.61 (all 1C, CH_2_*C*H_2_*C*H_2_*C*H_2_); 42.73, 43.37, 49.05 and 56.25 (*C*H_2_CH_2_CH_2_*C*H_2_ and *C*H_2_(cycle)); 121.37, 126.79, 129.14 and 143.50 (all 1C, *thiophene*).

MS-ESI: (+) 268.3 ([M + H]^+^, calc. 268.2).

**11c**: Starting from 0.4 g (0.8 mmol) of **10c**, product **11c**·*x*HCl was obtained in a yield of 0.28 g.

NMR (D_2_O): ^1^H: *δ* 1.28–1.39 (m, 2H, CH_2_CH_2_C*H*_2_CH_2_CH_2_); 1.60–1.71 (m, 6H, C*H*_2_C*H*_2_CH_2_C*H*_2_CH_2_); 2.67 (t, ^3^*J*_HH_ = 7.4, 2H, CH_2_CH_2_CH_2_CH_2_C*H*_2_); 3.22 (dd, ^3^*J*_HH_ = 6.8 and 4.7, 4H, C*H*_2_(cycle)); 3.32 (dd, ^3^*J*_HH_ = 6.8 and 4.7, 4H, C*H*_2_(cycle)); 3.42 (s, 4H, C*H*_2_(cycle)); 7.06 (dd, ^3^*J*_HH_ = 4.9, ^4^*J*_HH_ = 1.2, 1H, SCHC*H*); 7.11 (dd, ^4^*J*_HH_ = 3.0 and 1.2, 1H, SC*H*C); 7.40 (dd, ^3^*J*_HH_ = 4.9, ^4^*J*_HH_ = 3.0, 1H, SC*H*CH). ^13^C{^1^H}: *δ* 24.64, 26.74, 30.09 and 30.12 (all 1C, CH_2_*C*H_2_*C*H_2_*C*H_2_*C*H_2_); 42.98, 43.40, 49.49 and 57.00 (*C*H_2_CH_2_CH_2_CH_2_*C*H_2_ and *C*H_2_(cycle)); 121.42, 126.81, 129.55 and 144.19 (all 1C, *thiophene*).

MS-ESI: (+) 282.4 ([M + H]^+^, calc. 282.2).

#### 4.2.17. 1-[ω-(Thiophene-3-yl)-alkyl]-4,7-bis(*iso*-propyl)-1,4,7-triazacyclononane (alkyl: **L14a**—propyl, **L14b**—butyl, **L14c**—pentyl)

General procedure: compound **11a**–**c**·*x*HCl and K_2_CO_3_ (excess) were suspended in MeCN (30 mL) and stirred at room temperature for 1 h in order to convert the hydrochloride into a free amine base. Then, *iso*-propyl bromide (5 equiv.) was added, and the mixture was heated to 60 °C and left stirring overnight. The reaction progress was followed by HPLC-MS, and the alkylating agent was occasionally added in order to maximise the yield of the reaction. After the alkylation was complete according to HPLC-MS, the mixture was filtered through S4 frit and evaporated. The residue was redissolved in MeCN and purified using flash chromatography on reverse-phase silica gel (GM1). Product-containing fractions were combined and evaporated, yielding final ligands **L14a**–**c**·*x*TFA as colourless or light-yellow oils.

**L14a**: Starting from 0.55 g of **11a**·*x*HCl, product **L14a**·*x*TFA was obtained in a yield of 0.46 g.

NMR (D_2_O): ^1^H: *δ* 1.35 (d, ^3^*J*_HH_ = 6.3, 12H, C*H*_3_); 1.89–1.99 (m, 2H, CH_2_C*H*_2_CH_2_); 2.68 (t, ^3^*J*_HH_ = 7.3, 2H, C*H*_2_CH_2_CH_2_); 2.84–2.92 (m, 2H, CH_2_CH_2_C*H*_2_); 3.06–3.45 (br m, 8H, C*H*_2_(cycle)); 3.51 (s, 4H, C*H*_2_(cycle)); 3.64 (sept, ^3^*J*_HH_ = 6.6, 2H, C*H*); 7.07 (dd, ^3^*J*_HH_ = 5.0, ^4^*J*_HH_ = 1.5, 1H, SCHC*H*C); 7.14 (dd, ^4^*J*_HH_ = 3.3 and 1.3, 1H, SC*H*C); 7.43 (dd, ^3^*J*_HH_ = 5.0, ^4^*J*_HH_ = 3.3, 1H, SC*H*CH). ^13^C{^1^H}: *δ* 16.07, 17.67 (4C, *C*H_3_); 25.04, 27.44, 46.07, 47.71, 48.84, 55.29 and 59.92 (all 1C, *C*H_2_(cycle), *C*H_2_*C*H_2_*C*H_2_ and *C*H); 121.65, 127.02, 128.93 and 142.45 (all 1C, *thiophene*).

MS-ESI: (+) 338.3 ([M +H]^+^, calc. 338.3).

**L14b**: Starting from 0.37 of **11b**·*x*HCl, product **L14b**·*x*TFA was obtained in a yield of 0.31 g.

NMR (D_2_O): ^1^H: *δ* 1.32 (d, ^3^*J*_HH_ = 6.6, 12H, C*H*_3_); 1.57–1.67 (m, 4H, CH_2_C*H*_2_C*H*_2_CH_2_); 2.70 (t, *J*_HH_ = 6.6, 2H, C*H*_2_CH_2_CH_2_CH_2_)); 2.86–2.95 (m, 2H, CH_2_CH_2_CH_2_C*H*_2_); 3.08–3.41 (br m, 8H, C*H*_2_(cycle)); 3.45 (s, 4H, C*H*_2_(cycle)); 3.59 (sept, ^3^*J*_HH_ = 6.6, 2H, C*H*); 7.06 (dd, ^3^*J*_HH_ = 4.9, ^4^*J*_HH_ = 1.3, 1H, SCHC*H*); 7.11 (dd, ^4^*J*_HH_ = 2.9 and 1.1, 1H, SC*H*C); 7.41 (dd, ^3^*J*_HH_ = 4.9, ^4^*J*_HH_ = 3.0, 1H, SCHC*H*). ^13^C{^1^H}: *δ* 16.30,17.55 (4C, *C*H_3_); 23.81, 27.63, 29.58, 46.19, 47.52, 49.30, 56.44 and 59.64 (all 1C, *C*H_2_(cycle), *C*H_2_*C*H_2_*C*H_2_*C*H_2_ and *C*H); 121.41, 126.79, 129.11 and 143.35 (all 1C, *thiophene*).

MS-ESI: 352.4 (+) ([M + H]^+^, calc. 352.3).

**L14c**: Starting from 0.42 of **11c**·*x*HCl, product **L14c**·*x*TFA was obtained in a yield of 0.34 g.

NMR (CD_3_OD): ^1^H: *δ* 1.30 (d, ^3^*J*_HH_ = 7.1, 12H, C*H*_3_); 1.55–1.67 (m, 4H, CH_2_C*H*_2_C*H*_2_CH_2_CH_2_); 2.62 (t, ^3^*J*_HH_ = 7.6, 2H, CH_2_CH_2_CH_2_C*H*_2_CH_2_), 2.83–2.91 (m, 4H, C*H*_2_CH_2_CH_2_CH_2_C*H*_2_); 3.09 (br s, 2H, C*H*_2_(cycle)); 3.17–3.39 (br m, 8H, C*H*_2_(cycle)); 3.39–3.54 (br m, 4H, C*H*_2_(cycle)); 3.59 (sept, ^3^*J*_HH_ = 6.6, 2H, C*H*); 6.91–6.94 (m, 2H, SCHC*H* and SC*H*C); 7.24 (d, ^3^*J*_HH_ = 4.2, 1H, SCHC*H*). ^13^C{^1^H}: *δ* 17.72 (4C, *C*H_3_); 27.74, 30.97, 31.39; 46.77, 47.19,49.28, 50.01 (all very br), 57.30; 121.02, 126.22, 129.09 and 143.83 (all 1C, *thiophene*).

MS-ESI: (+) 366.4 ([M + H]^+^, calc. 366.3).

### 4.3. Preparation of Stock Solutions of the Cu(II)–L Complexes

Stock solutions of the Cu(II)–L complexes with *c* = 15 mM were prepared using solutions of CuCl_2_∙2H_2_O or Cu(OAc)_2_ and a slight excess (~10%) of the corresponding ligand, to assure the full complexation of the metal ion, the pH of the complex stock solution was adjusted to 8.5 to assure that the ligand was truly present in excess (if not, Cu(OH)_2_ precipitated; in such an occasional case, the ligand concentration was re-evaluated, and a new sample of the complex was prepared). The formation of the complexes was evidenced by a change in colour to deep blue or green (Appendix A) and via MS (Appendix A).

### 4.4. The Hydrolysis of BNPP, Followed by UV-Vis Spectroscopy

The hydrolytic experiments were carried out as follows in a 1 cm quartz cuvette at 37 °C (50 °C in specified cases) and at a pH of 7.5. The concentration of BNPP in the cuvette was 0.050 mM, and the concentrations of individual Cu(II) complexes were each 5 mM. The ionic strength of the reaction mixture was kept at 150 mM using NaCl, and the solution was buffered using HEPES buffer (50 mM). The samples were prepared using the following procedure: to a 1mL cuvette, 500 μL of HEPES (100 mM)/NaCl (300 mM) stock solution was added to buffer the solution to a pH of 7.5, and 167 μL of BNPP (0.300 mM) was pipetted. Right before the start of the experiment, 333 μL of a stock solution of Cu(II)–**L** complex (*c* = 15 mM) was added. A blank sample was prepared by mixing stock solutions of BNPP and HEPES(NaCl), and the complex solution was exchanged for water. The course of the reaction was monitored via the spectral change using a UV-Vis spectrophotometer in the range 350–900 nm. Spectra were acquired in 10 min or 20 s intervals over 24 h. The observed rate constants of BNPP hydrolysis were fitted from the time dependence of the absorption at 400 nm. The observed rate constants were evaluated either by fitting the data with an exponential decay equation or as the slope of the linear part of the dependence (for the slower reactions); see ESI.

Experiments testing the possible role of a free copper(II) ion in hydrolysis were performed analogously using a stock solution of CuCl_2_∙2H_2_O (*c* = 15 mM) instead of the Cu(II)–**L** complex. However, in this case, the pH was only 6.0 as the precipitation of Cu(OH)_2_ occurred at higher pH values. The possible spontaneous hydrolysis of BNPP at a pH of 7.5 was also tested. In both of these experiments, no hydrolysis of BNPP was observed.

For the experiment with an excess of substrate to complex (10:1 BNPP:Cu(II)–**L14b** ratio), the sample was prepared by the following way: to a 1mL cuvette, 500 μL of stock solution of HEPES (100 mM)/NaCl (300 mM) to buffer the solution to a pH of 7.5, 100 μL of BNPP (1 mM), 100 μL of Cu(II)–**L14b** (0.1 mM) and 100 μL of water were pipetted. A blank sample was prepared by mixing 500 μL of HEPES/NaCl solution, 100 μL BNPP (1 mM) and 400 μL water. The experiment was conducted at 50 °C for 5 days with 5 min measurement intervals.

### 4.5. The Hydrolysis of BNPP, Followed by HPLC

The metal complex used in this experiment needed to be removed before the HPLC was conducted as it absorbs strongly. Therefore, the experiment was modified when compared to procedure used for the UV-Vis experiment described above: 3 mL of stock solution of the Cu(II)–**L1** complex (24 mM) were mixed with 3 mL of NH_4_HCO_3_ (30 mM) as a buffer, and the pH was adjusted to 7.5 using diluted HCO_2_H. The mixture was tempered to 37 °C, and hydrolysis was initiated via the addition of 3 mL of BNPP (*c* = 9 mM). In regular intervals, 1 mL of the reaction mixture was poured on top of a column containing 1 mL of strong cation exchange resin in H^+^-form (Dowex 50), and the column was eluted with 4 mL of water. The aqueous eluate was analysed via HPLC (Waters Arc HPLC system, Cortecs C18 2.7 µm 4.6 × 50 mm column, isocratic elution, mobile phase 20mM NH_4_HCO_3_:MeOH 1:1, flow rate 1.2 mL∙min^−1^, absorbance detection at 330 nm; the wavelength was chosen to detect all three compounds whose absorption maxima differ significantly, see below). The individual compounds were identified via their UV spectra, which were measured by the detector (absorption maxima: BNPP 288 nm, NPP 312 nm, NP 403 nm), and their mass spectra (the most intensive peaks found in negative mode corresponded to mono-deprotonated forms of the compounds, [M−H]^−^: BNPP 338.9 (calc. 339.0), NPP 218.9 (calc. 219.0) and NP 137.9 (calc. 138.0)).

## 5. Conclusions

A set of novel derivatives of 1,4,7-triazacyclononane with a thiazole or thiophene side group was prepared; the side group affords the possibility of anchoring these derivatives into a polymeric material. The catalytic activity of the corresponding copper(II) complexes for the hydrolysis of phosphate ester bonds was studied using bis(*p*-nitrophenyl)phosphate as a model substrate. The influence of the linker’s length and its nature (a C_3_-to-C_5_ purely aliphatic chain or an amide-bond-containing chain) on the catalytic activity was studied. In general, C_3_-linkers with the possibility to close a six-membered chelate ring to the central Cu(II) ion efficiently block catalytic activity, but C_4_ and longer linkers can be utilised in efficient systems. The thiazole–carboxamide anchor probably partly binds the metal ion in a chelate ring and decreases catalytic efficiency. On the other hand, the thiophene anchor was found to not interfere with the catalytic centre, and the corresponding complexes belong to some of the most effective compounds ever tested for this purpose. Therefore, these novel, potentially bifunctional systems could provide the possibility of creating new coating materials for medicinal devices which would prevent biofilm formation.

## Data Availability

Data are contained within the article and Appendix A.

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
