# Peer review of "The Hydrolytic Activity of Copper(II) Complexes with 1,4,7-Triazacyclononane Derivatives for the Hydrolysis of Phosphate Diesters"

_molecules, 2023, doi:10.3390/molecules28227542_

Round 1

Reviewer 1 Report

Comments and Suggestions for Authors

Buzikava et. al. have worked on the manuscript titled “Catalytic activity of copper(II) complexes with 1,4,7-triazacy-2 clononane derivatives for phosphate diester hydrolysis”. The work has been performed well with due consideration of the catalyst activity using these Cu(II) based complexes.

The work can be published in the molecules. However the following issues to be resolved prior to final publication.

1.       There are some typos errors in the manuscripts., which need to be corrected.

2.       Figure 3B please increase the font for fitting parameters.

3.       Authors should provide NMR spectra and mass spectra (MS-ESI) in the supporting information.

4.       UV graphs indicating the change in the absorption intensity should be provide along with the calculations for rate constant. 

Comments on the Quality of English Language

Moderate editing of English language required

Reviewer 2 Report

Comments and Suggestions for Authors

In the manuscript entitled  "Catalytic activity of copper (II) complexes with 1,4,7-triazacy- 2 clononane derivatives for phosphate diester hydrolysis" by Kotek and coworkers is a nice piece of work and is recommended for favor of publication subject to following changes:  

1. Please provide FT-IR data of all the compounds and compare with previously reported literature.  

2.  Preparation of stock solutions of the Cu(II)–L complexes, here, why pH of the complex stock solution was adjusted to 8.5 ?  

3. Please provide TGA analysis of compounds and discuss stability.  

4. Please provide digital images and SEM-EDX study or XPS data of compound.  

5. For introduction: New functional complex materials should be cited and discussed appropriately, such as such as, e.g. (1) Cryst. Growth Des. 2022, 22, 7374−7394.  (2) Journal of Molecular Structure 1294 (2023) 136371.   

6. If possible, please draw the possible mechanism diagram for Catalytic hydrolysis of BNPP followed by HPLC and Catalytic hydrolysis of BNPP followed by UV-vis spectroscopy  

7. Please check and polish the paper carefully before the submission of the revised MS.  

8. The catalytic activity of the corresponding copper (II) complexes for hydrolysis of phosphate ester bond was studied using bis(p-nitro- phenyl)phosphate as a model substrate. Please check the reusability and stability of complexes and discuss in detail.

Comments on the Quality of English Language

 Minor editing of English language required

Reviewer 3 Report

Comments and Suggestions for Authors

The authors have reported a study on the catalytic activity of a series of Cu-TACN complexes. The synthesis and purification of the ligands have been conducted with care and are well-documented. However, there are concerns regarding the experimental methods and evaluation of the catalytic reactions.

1.     In Table 1, it is mentioned that the concentration of CuL and BNPP in the cuvette are 5 mM and 0.05 mM, respectively. Under these reaction conditions, the copper complex is present in 100 times the concentration of the substrate. It is unclear whether this constitutes a catalytic reaction. If the title includes "catalytic activity," you should reconsider the experimental conditions and provide evidence, such as turnover numbers, to demonstrate that it is indeed a catalytic reaction.

2.     The d-d absorption band of the copper complex appearing at 660 nm in Figure 3 appears to exhibit a slight time-dependent change. If it is decreasing, it could be considered that the complex is not acting as a catalyst but rather undergoing decomposition.

3.     There are errors in the units of the solution concentrations mentioned in Section4.4. Please correct these to the appropriate units. The exact correction for the units would depend on the context and the intended units (e.g., mM or M).

4.     In the NMR peak assignments provided from pages 9 to 17, there are numerous instances where "C" and "H" are in italics. Please make the necessary corrections.

5.     The term "bifunctional" is used throughout the paper, but there is no evidence to confirm whether the reported ligands actually function as bifunctional entities. If such confirmation is lacking, the descriptions from at least page 4 onwards should be removed.

Round 2

Reviewer 1 Report

Comments and Suggestions for Authors

The paper can be accepted in the present. The authors have made all the required changes requested by reviewers. 

Reviewer 3 Report

Comments and Suggestions for Authors

The paper can be published in the present form.